# Comprehensive Equilibrium Function Tests for an Accurate Diagnosis in Vertigo: A Retrospective Analysis

**DOI:** 10.3390/jcm13092450

**Published:** 2024-04-23

**Authors:** Shumpei Futami, Toru Miwa

**Affiliations:** 1Department of Otolaryngology, Osaka Metropolitan University, 1-4-3 Asahi-machi, Abeno-ku, Osaka 545-8585, Japan; futami@ent-ocu.com; 2Department of Otolaryngology-Head and Neck Surgery, Graduate of School of Medicine, Kyoto University, 54 Shogoin Kawahara-cho, Sakyo-ku, Kyoto 606-8507, Japan

**Keywords:** vertigo, equilibrium function test, diagnosis, multivariate logistic regression analysis

## Abstract

**Background/Objectives**: An accurate diagnosis of vertigo is crucial in patient care. Traditional balance function tests often fail to offer independent, conclusive diagnoses. This study aimed to bridge the gap between traditional diagnostic approaches and the evolving landscape of automated diagnostic tools, laying the groundwork for advancements in vertigo care. **Methods**: A cohort of 1400 individuals with dizziness underwent a battery of equilibrium function tests, and diagnoses were established based on the criteria by the Japanese Society for Vertigo and Equilibrium. A multivariate analysis identified the key diagnostic factors for various vestibudata nlar disorders, including Meniere’s disease, vestibular neuritis, and benign paroxysmal positional vertigo. **Results**: This study underscored the complexity of diagnosing certain disorders such as benign paroxysmal positional vertigo, where clinical symptoms play a crucial role. Additionally, it highlighted the utility of specific physical balance function tests for differentiating central diseases. These findings bolster the reliability of established diagnostic tools, such as audiometry for Meniere’s disease and spontaneous nystagmus for vestibular neuritis. **Conclusions**: This study concluded that a multifaceted approach integrating multiple diagnostic indicators is crucial for accurate clinical decisions in vestibular disorders. Future studies should incorporate novel tests, quantitative assessments, and advanced technologies to enhance the diagnostic capabilities of vestibular medicine.

## 1. Introduction

An accurate diagnosis of vertigo is paramount in patient care. While interviews play a crucial role, employing balance function tests is essential. However, despite their utility, most existing balance function tests often fail to yield a definitive diagnosis independently [1]. Considerable research has delved into the specific tests required for diagnosing various diseases, and profound dissimilarities from normal individuals have been observed in numerous studies. However, a comprehensive endeavor to identify universally valuable tests across diverse diseases remains conspicuously lacking. The existing literature underscores the importance of understanding the nuanced distinctions between disease-specific diagnostic tests and those applicable to healthy individuals [2,3,4,5], shedding light on the need for more tailored approaches. Despite these advancements, the challenge of consolidating diverse diagnostic tests within the constraints of daily medical practice persists. The sheer multitude of tests, each providing unique insights, makes comprehensive evaluation within a limited timeframe a formidable task. Recently, the landscape of medical diagnostics has witnessed a paradigm shift with the emergence of automated diagnostic tools [6,7]. This includes the promising integration of large-scale language models and deep-learning methodologies [6]. As the field evolves, the potential for automated diagnostic tools to revolutionize vertigo treatment is gradually gaining recognition [6,8]. This may allow quicker diagnosis in the future than that with conventional diagnostic approaches. Anticipating further strides in this direction, conducting a preliminary assessment of the suitability of existing data for automated diagnosis has become imperative [6]. This inquiry is crucial for ascertaining the applicability and reliability of current data in the development of automated systems tailored for effective vertigo diagnosis and treatment planning. This study aimed to bridge the gap between traditional diagnostic approaches and the evolution of automated diagnostic tools to perform useful diagnostic tests in a limited amount of time, reducing the burden on medical staffs and patients, and paving the way for future advances in vertigo treatment.

## 2. Materials and Methods

### 2.1. Participants

This retrospective cohort study comprised 1400 patients who underwent balance function tests at our hospital between April 1996 and March 1998. All patients had sought medical attention due to prevalent symptoms of dizziness and light-headedness within the specified timeframe. To uphold the integrity of our dataset, we implemented stringent exclusion criteria, meticulously screening and excluded patients with unclear or indistinguishable written records. This meticulous process was imperative to guarantee the reliability of our dataset, fortifying the foundation upon which our subsequent analyses and findings are built.

### 2.2. Equilibrium Function Tests and Evaluation

All the equilibrium function tests adhered to the stringent criteria proposed by the Japanese Society of Vertigo and Equilibrium Medicine. A comprehensive battery of static and dynamic body balance tests, including the Mann test, single-leg stance test, and stepping test, was conducted. Nystagmus tests, encompassing spontaneous and positional nystagmus, and positional change in nystagmus [9], were performed. Additional assessments included the eye tracking test (ETT) [9], optokinetic nystagmus (OKN) [9], and the caloric test [3]. The Schellong test [10], utilizing a manual sphygmomanometer, was employed for diagnosing orthostatic dysregulation. Furthermore, the Cornell Medical Index (CMI) and a standard pure-tone audiometric test provided valuable insights into the psychological and auditory dimensions [11].

#### 2.2.1. Mann Test

The patient stood upright with both feet in a straight line, with the heels touching the apex of the feet, and distributing the weight equally on both feet. Both upper limbs were placed lightly to the side of the body, the head position was maintained correctly, and the patient was asked to look straight ahead. Both eyes were observed while open and closed for 30 s each. The legs were placed in front and behind the patient, alternating from left to right.

#### 2.2.2. Single-Leg Stance Test

The patient stood upright on one leg with the other thigh nearly flat. The patient was observed for 30 s to determine the inability to fall. The patient was observed standing upright on the right and left leg with eyes open and closed.

#### 2.2.3. Stepping Test

The examinee stood in the center of a concentric circle without footwear with the front of his/her face and gaze fixed in the front, both upper limbs extended forward, palms facing downwards, and feet together. The exercises were performed by raising the thighs horizontally with open eyes and lightly stepping on the spot. Then, with the eyes shielded (closed), the same was performed for 100 steps. The patient’s posture (swaying and falling) was observed during the footsteps, and the rotation angle, transition angle, and transition distance trajectory at the patient’s stopping position after the footsteps were completed, measured, and recorded.

#### 2.2.4. ETT

The eye movement was recorded using electronystagmography. The subject’s head was fixed and the eyes tracked a smoothly moving optotype in front of the eye. The subject may not be able to follow the optotype smoothly if it was followed mechanically, so the subject was asked to follow the optotype with appropriate breaks in between. Stimulation methods include triangular waves with constant velocity stimulation, sinusoidal waves with ever-changing velocity, and circular ETT, which induces horizontal and vertical components simultaneously by making the subject perform circular movements. To eliminate the subject’s prediction, the optotype moves once in one direction, and the speed and direction of movement can be controlled randomly. The relationship between amplitude, frequency, and optokinetic velocity is (optokinetic velocity/amplitude) × 2 = frequency. Constant velocity stimuli are suitable for judging speed. Sinusoidal waves are suitable for observing the response pattern of eye movements to visual stimuli.

#### 2.2.5. OKN

After the ETT, the camera accelerated for 40–45 s to a final angular velocity of 160–180°/s at an isometric acceleration of 4°/s^2^ and then decelerated to a stop at 4°/s^2^. The nystagmus was recorded by electronystagmography at a paper feed rate of 1 mm/s.

#### 2.2.6. Caloric Test

The subject was placed in a supine position with the upper half of the body on a 30° slope or a pillow so that the lateral semicircular canal was vertical. Water was injected into the eyes at the prescribed volume and time using the respective water injection equipment. When the nystagmus was recorded using electronystagmography, the maximum slow phase velocity of the nystagmus was determined. The subjects were injected alternately with warm and cold water at body temperatures of plus and minus 7 °C (44 °C, 30 °C), and CP (canal paresis) was determined from the four responses.

#### 2.2.7. Schellong Test

Non-invasive oscillatory measurements of blood pressure (BP) and heart rate (HR) were recorded: (1) after 5 min in the supine position; (2) after 1 min standing; and (3) after 10 min standing. The cuff of the BP-recording device was attached to the left arm, which was supported at the heart level throughout the study. The testing was conducted during the daytime, in a quiet environment, at a constant room temperature of 22 °C–25 °C to exclude the effects of chronobiologic factors on the outcomes of the test. The participants maintained a regular meal schedule but were restricted from smoking and caffeine ingestion for 6 h before the examination. The intake of foods and medications with sympathomimetic activity was also prohibited before the study.

#### 2.2.8. CMI

It was a self-administered questionnaire that collected a large body of significant medical and psychiatric data without the physician’s participation so that the physicians have information on which tentative diagnostic appraisals of the patient’s total medical problem may be based, even before they interview the patient. This communication reported the accuracy and completeness of the diagnostic appraisals that can be made with the data on the CMI. CMI was a four-page, letter-size sheet on which 195 questions were printed corresponding closely to those usually asked in a detailed and comprehensive medical interview. Many questions on the psychological aspects of medical disorders were included. Questions were in informal language and worded to be understood by persons with a reading knowledge of simple English. We used the Japanese version of CMI.

### 2.3. Examination of Normality and Abnormalities

All conducted tests, including the Mann test, a single-leg stance test, stepping test, spontaneous nystagmus, positional nystagmus, positional change nystagmus, ETT, OKN, caloric test, Schellong test, CMI, and audiograms, were assessed for normality and abnormalities with reference to established criteria [12].

Positive judgment criteria for vestibular tests were briefly described as follows.

#### 2.3.1. Mann Test

Abnormality was identified when a fall occurs within 30 s of both eyes opening and closing, providing a distinct marker for compromised vestibular function. For a single-leg stance test, abnormality was identified when the raised foot contacted the ground within 15 s of closing the eyes during the single-leg standing phase, offering a precise measure of postural instability.

#### 2.3.2. Stepping Test

Abnormality was defined with the execution of 100 steps of foot-stamping, coupled with a rotation angle of 91° or more, introducing a quantitative metric for dynamic vestibular assessment. Abnormality was objectively quantified as a transition distance of 1 m or more, providing a clear threshold for assessing spatial disorientation.

#### 2.3.3. ETT

Abnormal saccadic patterns and ataxic saccadic patterns during ETT serve as robust positive indicators of vestibular dysfunction, enhancing the diagnostic precision of eye movement assessments.

#### 2.3.4. OKN

Abnormality was measured using multiple parameters, including nystagmus inversion, decreased slow phase velocity of nystagmus, reduced nystagmus count, rapid eye movements in the slow phase of nystagmus, and left-right differences in response, enriching the characterization of vestibular anomalies.

#### 2.3.5. Schellong Test

Abnormal findings comprise distinct markers, including a pulse pressure difference of 16 mmHg or more, a systolic pressure difference of 21 mmHg or more, and a maximum pulse rate difference of 21/min or more. These criteria offer comprehensive insights into orthostatic cardiovascular responses associated with vestibular dysfunction.

#### 2.3.6. CMI

It was meticulously evaluated on a graded scale from area I to IV, with Regions II to IV identified as the CMI high group, providing a nuanced classification for the severity of subjective symptoms related to vestibular dysfunction.

#### 2.3.7. Audiogram

An abnormal audiogram was distinctly defined by a hearing loss of 30 dB or more across the sextet of frequencies ranging from 250 to 8000 Hz, establishing a clear benchmark for auditory abnormalities associated with vestibular dysfunction. Audiometry was demonstrated using an audiometer (AA-75, RION Co., Ltd., Tokyo, Japan), and the above six frequencies were measured.

#### 2.3.8. Caloric Test

Abnormal findings encompass criteria for suspected CP, moderate CP, and severe CP, each contributing to a refined classification of vestibular dysfunction based on maximum slow phase velocity, duration, and responsiveness to 20 °C stimulation.

### 2.4. Diagnosis

Diagnoses were established based on the discerning criteria of the Japanese Society for Vertigo and Equilibrium at the time of examination [12]. The spectrum of diagnoses encompassed Meniere’s disease (MD), vestibular neuritis (VN), benign paroxysmal positional vertigo (BPPV), orthostatic dysregulation (OD), auditory tumor (AT), brain tumor, degenerative diseases, psychogenic vertigo, peripheral vestibular disorders, drug-induced vertigo, and Hunt syndrome.

#### 2.4.1. MD

When the following conditions (1)–(4) are present, MD is suspected (90%): (1) paroxysmal rotatory vertigo that recurs; (2) cochlear symptoms that fluctuate with attacks of vertigo; (3) absence of neurologic symptoms other than those of the eighth cranial nerve; (4) inability to identify the cause; (5) hearing loss characteristic of MD is present on audiometry; (6) balance function tests show evidence of inner ear damage; (7) neurological examinations show no damage other than to the eighth cranial nerve associated with vertigo; and (8) otorhinolaryngological, endoscopic, and laboratory tests show no cause of inner ear damage. If MD is suspected in cases with (1)–(4) and (5)–(8) are present on examination, MD is certain.

#### 2.4.2. VN

It is suspected when the conditions (1)–(4) are present: (1) sudden attacks of vertigo are the main complaint, most often a single major attack; (2) a feeling of lightheadedness or heaviness persists after an attack of vertigo; (3) no cochlear symptoms directly related to vertigo; (4) no history of any disease that may cause or precipitate vertigo; (5) upper respiratory tract infection around 7 to 10 days preceding the onset of vertigo; (6) caloric test shows a decreased or no temperature response on the affected side; (7) spontaneous and cephalic nystagmus tests show directionally fixed horizontal nystagmus during attacks of vertigo, usually in the healthy side; and (8) neurological examination reveals no evidence of neuropathy other than the vestibular nerve. If VN is suspected in cases with (1)–(4) and (6)–(8) are present on examination, VN is certain.

#### 2.4.3. BPPV

The diagnosis of “suspected BPPV” is made when (1)–(3) are present: (1) rotational or dynamic vertigo occurs when the patient assumes a specific head position (vertigo head position); (2) vertigo gradually increases in the vertigo head position, then decreases or disappears; (3) vertigo lightens or disappears when the patient continues in the same head position; and (4) hearing loss, tinnitus, and lightheadedness are often unnoticed. The patient was placed under Frenzel glasses and asked to change head positions from supine to left or right lateral recumbency, or from seated to suspended head position, and was examined for nystagmus and vertigo. The nystagmus characteristic of this syndrome often appears when the patient is moved from a sitting position to a suspended head position. (5) In the dizzy head position, nystagmus appears with a latency of a few seconds, gradually increases, and then diminishes or disappears. The patient is aware of dizziness with the onset of nystagmus. (6) When the patient is subsequently placed in the dizzy head position, the appearance of nystagmus and vertigo is clearly attenuated. (7) When the patient is returned to a sitting or supine position from the dizzy head position, a nystagmus, mainly rotatory, in the opposite direction may occur. (8) There are often no abnormal findings on hearing or caloric tests. (9) There are no directly related central nervous system symptoms. When the symptoms (6)–(8) are present, the diagnosis of BPPV is made.

#### 2.4.4. OD

OD is suspected when the following conditions (1)–(4) are met. (1) Non-rotatory vertigo is more common. (2) The duration of vertigo is relatively short. (3) It is often accompanied by complaints such as a feeling of inversion, stiff shoulders, tinnitus, or heaviness in the head. (4) It should be noted that hypertension, hypotension, orthostatic hypotension (orthostatic ataxia), arteriosclerosis, anemia, cardiac diseases such as arrhythmia, elderly patients, and drug users may be associated with dizziness. (5) Vertigo is often accompanied by hearing loss. (6) Blood pressure is often elevated during attacks of vertigo. (7) Hypertension, especially borderline hypertension, the presence of hypotension, and fluctuations in blood pressure and heart rate are observed, for example, by conducting daily examinations using the Schellong test; and daily blood pressure recordings of blood pressure, blood pressure circadian rhythm abnormalities, and heart rate variability. (8) Measuring autonomic nervous system function (systemic) includes the assessment of R-R interval variability, atropine, and heart rate variability with intravenous propranolol to examine abnormalities in both sympathetic and parasympathetic nervous system function. (9) Cervical sympathetic and parasympathetic nervous system function. (10) Vertebral artery blood flow examination: abnormal blood flow; left–right difference by Doppler ultrasound and digital subtraction angiography. (11) CT scan for abnormal findings. (12) Investigation of environmental stressors and psychological examination. The diagnosis of OD was confirmed when there is a high degree of suspicion by examining (7) and abnormalities by examining (8)–(10).

#### 2.4.5. AT, Brain Tumor, Degenerative Diseases

AT, brain tumors, and degenerative diseases were diagnosed using high-resolution contrast-enhanced magnetic resonance imaging.

#### 2.4.6. Psychogenic Vertigo

Psychogenic vertigo was diagnosed when the following three conditions were met: (1) other organic or functional vertigo was excluded, (2) there was a psychogenic trigger for the onset of vertigo, and (3) the results and responses to balance function tests and questionnaires were not reproducible.

#### 2.4.7. Peripheral Vestibular Disorder

The diagnosis of peripheral vestibular disorder was based on the following criteria: (1) no evidence of central nervous system abnormality; (2) dizziness, mainly rotatory vertigo; and (3) other peripheral vestibular disorder diseases or causes could be ruled out.

#### 2.4.8. Drug-Induced Vertigo

The diagnosis of drug-induced vertigo was defined as meeting the following criteria: (1) occurring during or immediately after drug administration, (2) dizziness mainly consisting of a floating sensation, (3) Jumbling phenomenon, and (4) no central nervous system abnormalities.

#### 2.4.9. Hunt Syndrome

The diagnosis of Hunt syndrome was based on the following criteria: (1) blistering and redness around the auricle, (2) cochlear symptoms, and (3) facial nerve palsy.

### 2.5. Statistical Analysis

To determine the diagnostic utility of the tests, logistic multivariate analysis was performed with MD, OD, VN, BPPV, AT, degenerative diseases, psychogenic vertigo, peripheral vestibular disorders, otitis media, drug-induced dizziness, brain tumors, and Hunt syndrome designated as objective variables. The explanatory variables include age, sex, Mann test, single-leg stance test, stepping test, nystagmus tests, ETT, OKN, caloric test, Schellong test, CMI, and audiograms. A model was created after confirming the variance inflation factor. The Akaike’s Information Criterion (AIC) was used to balance the complexity of the model with the goodness of fit to the data. Missing values were addressed using forest plots, and non-convergent data were excluded from the analysis. The evaluation results were considered not applicable if the calculated sample size after data collection was insufficient for statistical analysis. Statistical analyses were conducted using GraphPad Prism (Ver 9.5.0; GraphPad Software, Boston, MA, USA) to ensure rigor and reliability.

## 3. Results

In total, 1392 patients were included in the study after excluding 8 with illegible prints. The mean age of the participants was 53.2 ± 15.8 years, comprising 489 men and 897 women. Multiple diagnoses were observed in 44% (*n* = 616) of the patients. The specific diagnoses included 134 cases of MD, 30 of VN, 138 of BPPV, 40 of otitis media, 83 of AT, 61 of degenerative diseases, 726 of OD, 74 of psychogenic dizziness, 490 of peripheral vestibular disorder, 8 of drug-induced dizziness, 4 of brain tumor, and 2 of Hunt syndrome.

Multivariate analysis results are summarized in Table 1.

### 3.1. MD

For the diagnosis of MD, audiograms were found to be significant (10.70 [3.03–37.70], *p* < 0.001).

### 3.2. OD

Items required for the diagnosis of OD included positional nystagmus (0.47 [0.29–0.77], *p* = 0.002), the Schellong test (420.00 [42.10–4190.00], *p* < 0.001), sex (2.59 [1.05–6.37], *p* < 0.04), and stepping test (5.84 [1.11–30.70], *p* < 0.04).

### 3.3. VN

The key diagnostic feature for VN was spontaneous nystagmus (5.90 [0.51–68.5]), although it was not statistically significant.

### 3.4. BPPV

Items crucial for the diagnosis of BPPV included CMI (2.72 [0.94–7.91]), spontaneous nystagmus (2.74 [0.96–7.88]), ETT (0.15 [0.02–1.19]), and the Schellong test (4.14 [0.88–19.6]); however, these differences were not statistically significant.

### 3.5. AT

AT diagnosis relied on single-leg stance test (2.95 [0.14–64.10]), stepping test (2.60 [0.37–18.40]), and head-turning nystagmus (2.45 [0.50–0.50]); however, it was not statistically significant.

### 3.6. Degenerative Diseases

Single-leg stance test (70.55 [3.03–1880.00], *p* = 0.01) emerged as the critical diagnostic factor for degenerative diseases.

### 3.7. Psychogenic Vertigo

For the diagnosis of psychogenic vertigo, age (1.05 [1.00–1.09]) and the Schellong test (0.285 [0.80–53.80]) were identified; however, the differences were not statistically significant.

### 3.8. Peripheral Vestibular Disorder

Single-leg stance test (5.74 [1.18–27.90], *p* = 0.03 *) was the pivotal item for diagnosing peripheral vestibular disorder.

### 3.9. Others

Cases of otitis media, drug-induced dizziness, brain tumors, and Hunt syndrome were insufficient for meaningful statistical analysis.

## 4. Discussion

Our study successfully identified key equilibrium function tests with diagnostic validity for various vestibular disorders, illuminating both the confirming and challenging aspects of established diagnostic methods. The identification of specific tests for disease diagnosis aligns with existing literature [13], confirming the reliability of specific diagnostic indicators. The association between audiometry and MD diagnosis, spontaneous nystagmus for VN, and the Schellong test for OD diagnosis from previous reports, enhances the consistency and reliability of these diagnostic tools [12,13].

The diagnostic profile of BPPV is complex. While directionally alternating nystagmus appeared crucial, the high odds ratios for spontaneous nystagmus, CMI, and OD indicated the significance of multiple factors. The lack of high odds ratios for positional nystagmus and positional change nystagmus indicates that clinical symptoms were crucial in BPPV diagnosis, despite the absence of nystagmus during equilibrium function testing [12]. This nuanced approach to diagnosing BPPV underscores the importance of considering clinical symptoms, particularly in suspected cases [12]. The timeframe from the initial visit to the equilibrium function test at a university hospital might have influenced these results, emphasizing the need for a comprehensive assessment [14].

Furthermore, our findings highlighted the utility of the same physical balance function test for peripheral vestibular disorders. The absence of detection of the caloric test as a factor for peripheral vestibular dysfunction suggests its limitations, as abnormalities in other diseases may lead to false positives [3]. This emphasizes the need for caution when relying solely on caloric testing for the diagnosis of peripheral vestibular dysfunction.

Intriguingly, AT and degenerative diseases are challenging to diagnose using equilibrium function tests alone. However, the importance of body equilibrium function tests, such as the single-leg stance test and stepping tests, which form part of the neurological findings, implies their utility in differentiating central diseases [2]. This insight is particularly valuable for primary care physicians without access to imaging tests, as these simple tests offer a practical means of distinguishing between diseases.

The high odds ratios for age and the Schellong test in psychogenic dizziness align with the well-established association between psychiatric disorders, age, and autonomic nervous system function [15,16,17,18]. This supports the rationality of these results; however, the absence of CMI as a significant factor underscores the need to consider its non-specificity for psychogenic vertigo. While the CMI is used in psychosomatic disorders and psychiatric conditions, its exclusion as a key factor in this study emphasizes the importance of a multifaceted approach to understanding psychogenic dizziness [15].

### Limitations

Our study had certain limitations, such as the exclusion of emerging conditions (persistent postural–perceptual dizziness [19] and vestibular migraine [20], which hinder a fully contemporary perspective owing to its retrospective design. Restrictions on traditional diagnostic tests omit quantitative evaluation. Future research should integrate advanced assessments (vestibular evoked myogenic potential, video head impulse test, and inner ear contrast-enhanced magnetic resonance imaging) to enhance our understanding [21,22,23]. The absence of AI and deep learning techniques in data analysis signals a potential area for advancement, promising more nuanced insights. Another limitation of this study was the large number of cases in which the diagnosis was made by history. Unfortunately, the medical history was not available. There have been several reports on the use of medical history data [24,25], and we believe that in the future, it will be necessary to use digitalized questionnaires to improve the diagnosis accuracy. Finally, a small number of vestibular diseases such as vertigo with otitis media, drug-induced dizziness, vertigo/dizziness with brain tumors, and vertigo due to Hunt syndrome cannot be statistically determined. Additionally, the present study was conducted at a single institution. Since this is a preliminary study conducted at a single institution, we believe that future studies should either include a large sample size or prospectively examine the data at a large number of institutions.

## 5. Conclusions

Our comprehensive study meticulously examined key equilibrium function test factors, including audiogram, nystagmus, Schellong test, single-leg stance test, and stepping test, all of which bear diagnostic relevance across various vestibular disorders. This exhaustive exploration illuminates the critical role of established diagnostic tests in clinical assessments, signifying their pivotal contribution to advancing knowledge within the field of vestibular disorders. The study compellingly underscores the inadequacy of relying on a singular test for definitive disease diagnosis, shedding light on the intricate nature of vestibular disorders. Emphasizing the necessity for a multifaceted approach, it highlights that a holistic analysis incorporating multiple indicators is paramount for making accurate clinical decisions. This nuanced understanding of factors influencing vestibular pathology is indispensable in navigating the complexity of diagnosis. Considering this as a future scope, our research is a preliminary investigation to fill the gap in the application of large-scale language models and deep learning methods to new dizziness diagnostic techniques and to advocate for the integration of new tests and quantitative assessments. This research forms the baseline in terms of its ultimate goal of reducing the burden on medical staff and patients. This forward-thinking approach is crucial to staying at the forefront of scientific innovation in the realm of vestibular medicine. Our unwavering commitment to scientific excellence is geared towards enhancing the diagnostic capabilities of vestibular medicine, with the ultimate goal of making impactful contributions to both clinical practice and the broader scientific understanding of vestibular disorders.

## Figures and Tables

**Table 1 jcm-13-02450-t001:** Multivariate logistic regression analysis.

Objective Variables	Explanatory Variables	Odds Ratio	95% CI	*p* Value
MD	(Intercept)	0.03	0.009–0.09	<0.001
	Audiogram	10.70	3.03–37.70	<0.001 ***
OD	(Intercept)	0.00	0.00–0.05	0.00
	Sex	2.59	1.05–6.37	0.04 *
	Stepping test	5.84	1.11–30.70	0.04 *
	Nys (Auto)	2.23	0.86–5.80	0.1
	Nys (positional)	0.47	0.29–0.77	0.002 **
	Nys (positional change)	2.23	0.86–5.81	0.1
	Schellong test	420.00	42.10–4190.00	<0.001 ***
VN	(Intercept)	3.47 × 10^−10^	0.00–Inf	1.00
	Nys (Auto)	5.90	0.51–68.50	0.16
BPPV	(Intercept)	0.02	0.00–0.13	0.002
	ETT	0.15	0.02–1.19	0.07
	CMI	2.72	0.94–7.91	0.07
	Nys (Auto)	2.74	0.96–7.88	0.06
	Schellong test	4.14	0.88–19.60	0.07
AT	(Intercept)	0.10	0.00–2.88	0.18
	Age	0.99	0.95–1.04	0.67
	Sex	1.20	0.31–4.65	0.79
	Audiogram	0.73	0.20–2.68	0.64
	Single-leg stance test	2.95	0.14–64.10	0.49
	Mann test	0.76	0.06–9.35	0.83
	Stepping test	2.60	0.37–18.40	0.34
	OKN	0.85	0.05–13.80	0.91
	Nys (Auto)	2.07	0.54–7.97	0.29
	Nys (positional)	0.57	0.20–1.60	0.28
	Nys (positional change)	2.45	0.50–0.50	0.27
	Caloric test	1.20	0.29–4.96	0.80
	Schellong test	0.41	0.10–1.66	0.21
	CMI	1.07	0.25–4.58	0.93
Degeneration disorder	(Intercept)	0.046	0.00977–0.22	0.00
	Audiogram	3.47	0.669–18.00	0.14
	Single-leg stance test	70.55	3.03–1880.00	0.01 **
	Schellong test	0.285	0.06–1.26	0.1
Psychogenic vertigo	(Intercept)	0.0014	0.00–0.05	0.00
	Age	1.05	1.00–1.09	0.06
	Schellong test	6.54	0.80–53.80	0.08
Peripheral vestibular disorders	(Intercept)	0.50	0.19–1.33	0.17
	Single-leg stance test	5.74	1.18–27.90	0.03 *
	Nys (positional)	0.71	0.49–1.05	0.09
	Nys (positional change)	1.74	0.89–3.42	0.11

MD, Ménière’s Disease; OD, orthostatic dysregulation; VN, vestibular neuritis; BPPV, benign paroxysmal positional vertigo; AT, acoustic tumor; Nys, nystagmus; ETT, eye tracking test; CMI, Cornell’s Medical Index; OKN, optokinetic nystagmus. *: *p* < 0.05, **: *p* < 0.01, ***: *p* < 0.001

## Data Availability

Upon publication of this manuscript, the de-identified participant data, study protocol, and statistical analysis plan will be available upon request from academic institutions upon receipt of a credible research proposal approved by the corresponding author. Documents will be available in English only for a pre-specified time (typically 12 months) on a password-protected portal.

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
