# Peer review of "Comprehensive Equilibrium Function Tests for an Accurate Diagnosis in Vertigo: A Retrospective Analysis"

_jcm, 2024, doi:10.3390/jcm13092450_

Round 1

Reviewer 1 Report

Comments and Suggestions for Authors

- What you have for the introduction is a good, but could use a hypothesis statement/description as well as more explicit description of what exactly the study aimed to do (e.g., to bridge the gap by... narrowing the field to the best tools? comparing the best tools/combinations of tools to reduce the burden on medical professionals?)

- Starting at line 73, each test should be preceded by "the" (e.g., the Mann test)

- Should each of the protocols for these tests be cited? 

- The description starting at line 118 - is this a description of how all the tests were conducted or just the Schellong test?

- Would it be possible to combine 2.2 and 2.3 and break them up by test instead? Meaning that the Mann test and its abnormality would be described, then the stepping test, etc.) 2.2 and 2.3 are kind of hard to read right now as they are significant blocks of text.

- The same would be true for 2.4. Could this information be put into a table? Or bullet points? Following the information in the current format is challenging.

- The results in the paragraph starting at 239 could be presented in a table.

- The structure of sections 3.1 - 3.9 would be good to apply to section 2.

- The conclusions section should be supported more by the introduction. If the information in the conclusions was backed by the introduction as well as the results, then the impact would be stronger.

-The discussion section is strong and the rest of the paper would benefit from a similar writing style. 

Comments on the Quality of English Language

Minor editing is required as described in the above section.

Author Response

- What you have for the introduction is a good, but could use a hypothesis statement/description as well as more explicit description of what exactly the study aimed to do (e.g., to bridge the gap by... narrowing the field to the best tools? comparing the best tools/combinations of tools to reduce the burden on medical professionals?)

>Thank you for your suggestion, I have added it at the end of the Introduction section.

“This study aimed to bridge the gap between traditional diagnostic approaches and the evolution of automated diagnostic tools to perform useful diagnostic tests in a limited amount of time, reducing the burden on medical staff and patients, and paving the way for future advances in vertigo treatment.”

- Starting at line 73, each test should be preceded by "the" (e.g., the Mann test)

>Thank you very much. We used numbers instead of "the" because we numbered for clarity, not in paragraph form.

- Should each of the protocols for these tests be cited? 

>As you pointed out, it may not be necessary, but it is included due to the 4000 words limit stipulated by the MDPI.

- The description starting at line 118 - is this a description of how all the tests were conducted or just the Schellong test?

>Thank you very much. This is the explanation for Schellong test only. As mentioned previously, we have numbered and sub-divided each section for better clarity.

- Would it be possible to combine 2.2 and 2.3 and break them up by test instead? Meaning that the Mann test and its abnormality would be described, then the stepping test, etc.) 2.2 and 2.3 are kind of hard to read right now as they are significant blocks of text.

> We apologize for the lack in clarity. We could not merge 2.2 and 2.3 because their descriptions do not match, but as you suggested, we changed the structure such that it is similar to sections 3.1 to 3.9.

- The same would be true for 2.4. Could this information be put into a table? Or bullet points? Following the information in the current format is challenging.

>The text was initially prepared as a table; however, due to the journal’s word count as mentioned above, we converted it to text form. We have made some changes to the structure for easier comprehension.

- The results in the paragraph starting at 239 could be presented in a table.

>We apologize for the lengthiness. However, as described previously, we have presented it in text form rather than a table to achieve the journal’s minimum word count of 4000 words.

- The structure of sections 3.1 - 3.9 would be good to apply to section 2.

>Thank you for your suggestion. We have changed it accordingly.

- The conclusions section should be supported more by the introduction. If the information in the conclusions was backed by the introduction as well as the results, then the impact would be stronger.

> Thank you for the suggestion. We have mentioned in the Introduction section that this is a preliminary study for a large-scale language model analysis and that the ultimate goal is to reduce the burden on medical staffs and patients in the Conclusion section.

“Considering as a future scope, our research is a preliminary investigation to fill a gap in the application of large-scale language models and deep learning methods to new dizziness diagnostic techniques and to advocate for the integration of new tests and quantitative assessments. This research forms the baseline in terms of its ultimate goal of reducing the burden on medical staff and patients.”

-The discussion section is strong and the rest of the paper would benefit from a similar writing style. 

>Thank you for the encouragement. We have revised the manuscript for better comprehension as much as possible considering the required minimum word count.

Reviewer 2 Report

Comments and Suggestions for Authors

The Journal Of Clinical Medicine JCM (ISSN 2077-0383)

Manuscript ID jcm-2925833

Title of the manuscript: Comprehensive Equilibrium Function Tests for an Accurate Diagnosis in Vertigo: A Retrospective Analysis

Dear Editor, Dear Authors

Thank you sincerely for granting me the opportunity to evaluate your work.

This study is a cohort study that included 1,400 people who experienced dizziness. These individuals conducted a series of tests to assess their balance function. The study aimed to close the disparity between conventional diagnostic methods and the developing field of automated diagnostic tools, facilitating future progress in treating vertigo.

Dizziness is significant in clinical practice, and accurately determining a diagnosis is vital for the patient. This interdisciplinary topic encompasses the fields of otolaryngology, neurology, and family medicine. Conducting a study on the collateralization of diagnostic tests used in diagnosing dizziness can provide significant benefits, including reducing the time required for patient evaluation.

The abstract is accurately composed. The article's introduction presents information on the topic. However, the concluding sentence, which states the paper's purpose, may confuse the reader. It claims that the study aims to connect traditional diagnostic methods with the developing field of automated diagnostic tools to facilitate future progress in vertigo care. The study summarises the utility of the tests but does not explicitly discuss their application with artificial intelligence (AI).

• Were the authors clear about their objectives?

The Materials and Methods section lacks a description of the study group. Did all of the patients come from one specific medical facility? What were the participants' ages, and was a single study methodology employed? Was the study conducted prospectively or retrospectively? What criteria determined which individuals were included or excluded from the study group?

Section 2.2 discusses the concept of an equilibrium function. The section on Tests and Evaluation provides a sequential description of the tests used, except the Cornell Medical Index (the description is missing). To provide a complete understanding of tonal audiometry, it is necessary to provide a detailed account of the instrument and equipment utilized and the spectrum of frequencies examined.

Section 2.3, titled "Examination of normality and abnormalities," outlines the standards used to assess test results. Therefore, it is recommended that lines 156 to 166 be relocated to the preceding section that delineates the testing methodologies employed.

Section 2.4 Diagnosis does not describe the diagnostic criteria for the following conditions: auditory tumour (AT), degenerative illnesses, psychogenic vertigo, peripheral vestibular disorders, drug-induced vertigo, brain tumour, and Hunt syndrome.

Verse 220 contains an erroneous test title.

The statistical tests used to construct the results have not been specified.

Did the estate analysis include the diagnoses of otitis media, drug-induced dizziness, brain tumour, and Hunt syndrome? Were they allocated to the groups outlined in Table 1? or omitted from the displayed outcomes?

The authors emphasized the limitations of their investigation.

The conclusions are overly generalized and do not entirely align with the abovementioned findings. The work is intellectually stimulating, although complex language, given the intricate subject matter, hampers understanding and appreciation. I suggest one of the authors review and edit it one more time.

Author Response

Thank you sincerely for granting me the opportunity to evaluate your work.

This study is a cohort study that included 1,400 people who experienced dizziness. These individuals conducted a series of tests to assess their balance function. The study aimed to close the disparity between conventional diagnostic methods and the developing field of automated diagnostic tools, facilitating future progress in treating vertigo.

Dizziness is significant in clinical practice, and accurately determining a diagnosis is vital for the patient. This interdisciplinary topic encompasses the fields of otolaryngology, neurology, and family medicine. Conducting a study on the collateralization of diagnostic tests used in diagnosing dizziness can provide significant benefits, including reducing the time required for patient evaluation.

The abstract is accurately composed. The article's introduction presents information on the topic. However, the concluding sentence, which states the paper's purpose, may confuse the reader. It claims that the study aims to connect traditional diagnostic methods with the developing field of automated diagnostic tools to facilitate future progress in vertigo care. The study summarises the utility of the tests but does not explicitly discuss their application with artificial intelligence (AI).

>Thank you for your comments. The primary objective of this research is to bridge the gap between conventional diagnostics and the field of automated diagnostic tools, rather than to link the two. We believe that the development of automated diagnostics requires the evaluation of conventional diagnostic methods and the accumulation of data. We have revised the concluding sentence of the Introduction section to convey the aim clearly.

  • Were the authors clear about their objectives?

The Materials and Methods section lacks a description of the study group. Did all of the patients come from one specific medical facility? What were the participants' ages, and was a single study methodology employed? Was the study conducted prospectively or retrospectively? What criteria determined which individuals were included or excluded from the study group?

>Thank you for the comment. This study was a retrospective study, including all participants who underwent equilibrium function scrutiny at our facility between April 1996 and March 1998. We have added this description under section 2.1. The age of the participants is described in the Results section.

Section 2.2 discusses the concept of an equilibrium function. The section on Tests and Evaluation provides a sequential description of the tests used, except the Cornell Medical Index (the description is missing). To provide a complete understanding of tonal audiometry, it is necessary to provide a detailed account of the instrument and equipment utilized and the spectrum of frequencies examined.

>Thank you for pointing it out, I have included it in Section 2.3.7.

“Audiometry was demonstrated using audiometer (AA-75, RION Co., Ltd., Tokyo, Ja-pan), and the above six frequencies were measured.”

Section 2.3, titled "Examination of normality and abnormalities," outlines the standards used to assess test results. Therefore, it is recommended that lines 156 to 166 be relocated to the preceding section that delineates the testing methodologies employed.

> Thank you for pointing this out. I have moved it to Section 2.2.8.

Section 2.4 Diagnosis does not describe the diagnostic criteria for the following conditions: auditory tumour (AT), degenerative illnesses, psychogenic vertigo, peripheral vestibular disorders, drug-induced vertigo, brain tumour, and Hunt syndrome.

> Thank you for pointing it out, I have added them in Section 2.4.5-9.

Verse 220 contains an erroneous test title.

>Thank you for pointing it out. I have revised it.

The statistical tests used to construct the results have not been specified.

>Thank you for your comments. Description of the statistical analysis was added in Section 2.5.

“To determine the diagnostic utility of the tests, logistic multivariate analysis was performed with MD, OD, VN, BPPV, AT, Degenerative diseases, Psychogenic vertigo, Peripheral vestibular disorders, otitis media, drug-induced dizziness, brain tumors, and Hunt syndrome designated as objective variables. The explanatory variables in-clude age, sex, Mann test, single-leg stance test, stepping test, nystagmus tests, ETT, OKN, caloric test, Schellong test, CMI, and audiograms. A model was created after confirming the variance inflation factor. The Akaike's Information Criterion (AIC) was used to balance the complexity of the model with the goodness of fit to the data. Miss-ing values were addressed using forest plots, and non-convergent data were excluded from the analysis. The evaluation results were considered not applicable if the calcu-lated sample size after data collection was insufficient for statistical analysis.”

Did the estate analysis include the diagnoses of otitis media, drug-induced dizziness, brain tumour, and Hunt syndrome? Were they allocated to the groups outlined in Table 1? or omitted from the displayed outcomes?

>Thank you for the comment. Statistical analysis was also performed for otitis media, drug-induced vertigo, brain tumor, and Hunt syndrome; however, the results were not obtained due to lack of convergence in the analysis. We excluded cases other than those with complications of the diseases listed in the table.

The authors emphasized the limitations of their investigation.

The conclusions are overly generalized and do not entirely align with the abovementioned findings. The work is intellectually stimulating, although complex language, given the intricate subject matter, hampers understanding and appreciation. I suggest one of the authors review and edit it one more time.

>Thank you for your insightful comment. The co-author has reviewed and revised the manuscript.

Reviewer 3 Report

Comments and Suggestions for Authors

This manuscript on comprehensive equilibrium function tests for vertigo diagnosis presents potentially valuable insights into the diagnostic process of vestibular disorders. The use of a large cohort and multivariate analysis strengthens your findings. However, there are areas for enhancement to further elevate the manuscript's contribution to clinical medicine.

Below are my specific recommendations:

·        Consider expanding on how these findings can be applied in clinical settings lacking advanced diagnostic tools

·        I would recommend including a brief discussion regarding the potential for integrating new technologies like AI and deep learning in future studies, reflecting on the limitations section

·        Please clarify the selection criteria for your patient cohort to ensure readers understand the study's applicability

·        I would detail the statistical methods more thoroughly to aid in the reproducibility of your results

Author Response

This manuscript on comprehensive equilibrium function tests for vertigo diagnosis presents potentially valuable insights into the diagnostic process of vestibular disorders. The use of a large cohort and multivariate analysis strengthens your findings. However, there are areas for enhancement to further elevate the manuscript's contribution to clinical medicine.

Below are my specific recommendations:

  • Consider expanding on how these findings can be applied in clinical settings lacking advanced diagnostic tools

>Thank you for the suggestion. In this study, we examined tests that are useful for diagnosis in vertigo treatment. We have added a note in the Introduction section on the possibility that this could reduce wastage of time and decrease the burden on patients and providers.

“This may allow quicker diagnosis in the future than that with conventional diagnostic approaches.”

  • I would recommend including a brief discussion regarding the potential for integrating new technologies like AI and deep learning in future studies, reflecting on the limitations section

>Thank you for your comments, I have addressed what I said in Limitation along with the future scope in the Conclusion section instead of the Discussion section.

We have emphasized that this is a preliminary study that bridges the gap between large-scale language models and deep learning.

  • Please clarify the selection criteria for your patient cohort to ensure readers understand the study's applicability

>Thank you for the suggestion. I have added it in Section 2.1.

  • I would detail the statistical methods more thoroughly to aid in the reproducibility of your results

> Thank you for the suggestion. I have described the statistical method so that the study is reproducible.

Round 2

Reviewer 1 Report

Comments and Suggestions for Authors

Thank you for making those revisions - the paper is much clearer! 

Reviewer 3 Report

Comments and Suggestions for Authors

Congratulations on the improvements made to your manuscript. The added details and clarifications have enhanced the clarity and depth of your study, making it a valuable contribution to the field of vestibular disorders.